# Short communication: Learning How Landscapes Evolve with Neural Operators

Gareth G. Roberts[1]

[1]Department of Earth Science and Engineering, Imperial College London, Royal School of Mines, South Kensington, London, SW7 2AZ, UK

**Correspondence:** Gareth G. Roberts (gareth.roberts@imperial.ac.uk)

**Abstract.** The use of Fourier Neural Operators (FNOs) to learn how landscapes evolve is introduced. The approach makes use of recent developments in *deep learning* to learn the processes involved in evolving landscapes (e.g. erosion). An example is provided in which FNOs are developed using input-output pairs (elevations at different times) in synthetic landscapes generated using the stream power model (SPM). The SPM takes the form of a non-linear partial differential equation that advects slopes headwards. The results indicate that the learned operators can reliably and very rapidly predict subsequent landscape evolution at large scales. These results suggest that FNOs could be used to rapidly predict landscape evolution without recourse to the (slow) computation of flow routing and time stepping needed when generating numerical solutions to the SPM. More broadly they suggest that neural operators could be used to learn the processes that evolve actual and analogue landscapes. Interesting future work could involve assessment of whether learned operators can be applied to other settings or model parameterizations.

## Short Summary

The use of new Artificial Intelligence (AI) techniques to learn how landscapes evolve is demonstrated. A few 'snapshots' of an eroding landscape at different stages of its history provide enough information for AI to ascertain rules governing its evolution. Once the rules are known, predicting landscape evolution is extremely rapid and efficient, providing new tools to understand landscape change.

## 1 Introduction

This paper addresses two challenges in geomorphology. First, a general one: development of landscape evolution 'laws' or 'rules'. The second concerns generating predictions of landscape evolution efficiently and rapidly. Doing so is important for establishing the processes (e.g. uplift, erosion, climate, biota) that play a role in generating landscapes. Efficient prediction of landscape geometries (e.g. elevation as a function of space and time) is central to the recovery of histories of such processes from observed landscapes via inverse modeling (see e.g. Roberts and White, 2010; Croissant and Braun, 2014; Goren et al., 2014; Glotzbach, 2015; Fernandes et al., 2019; Barnhart et al., 2020). I explore the use of recently developed Fourier Neural Operators to address these challenges (Li et al., 2022; Kovachki et al., 2023).

Understanding how landscapes evolve is a cornerstone of geomorphology and provides useful information for many problems in geology and paleobiology, as well as for hazard and resource assessment (see e.g. Anderson and Anderson, 2010; Fernandes et al., 2019; Perrigo et al., 2020; Hoggard et al., 2021; Turner et al., 2023, and references therein). A variety of approaches exist to predict how they evolve in response to tectonic, climatic and other forcings. These include physical experimentation, e.g. at the scale of flume tanks, and field observations (see e.g. Bonnet and Crave, 2003; Scheingross et al., 2017, and references therein). They also include phenomenological and physics-based deterministic and stochastic landscape evolution models (LEMs). Such models are used to predict landscape evolution from erosional 'atomistic' scales, e.g. $< 1$ m and $< 1$ s, up to the largest scales, e.g. continents and tens of millions of years (see e.g. Hobley et al., 2017; Roberts and Wani, 2024, and references therein). Such models can be developed by combining observations and physics-based insights across scales of interest, and calibrated with independent geological and geophysical information (see e.g. Anderson and Anderson, 2010; Lague, 2014; Fernandes et al., 2019; Roberts and Wani, 2024, and references therein)

In contrast, the neural operator approach seeks to learn the mapping between function spaces from observations. In our case, the function spaces are landscapes at different times and the mapping could be regarded as learning, say, the erosional processes that evolved the landscape. In other words, we seek to answer the following question: What is the operation that has occurred to convert (evolve) a landscape from one time to another? So, instead of assuming that we know the erosional processes responsible for evolving a landscape, for instance, we seek to learn them from the information available, e.g. a landscape at different stages of its evolution. Similar questions have been addressed in other branches of the physical sciences. For instance, Fourier Neural Operators have been used to learn mappings between function spaces generated by solutions to partial differential equations (PDEs) including Burgers', Darcy flow and Navier-Stokes (see e.g. Li et al., 2022). Physics Informed Neural Operators (PINOs) have been developed to combine training data (e.g. input-output pairs) and constraints from physics to learn solution operators for partial differential equations (see e.g. Li et al., 2024).

Despite knowing modern topography very well (from satellite-derived measurements for instance), developing neural operators using actual landscapes is a very difficult problem because we do not usually know their histories (previous function spaces) with much precision. In contrast, realistic looking 'landscapes' have been produced in flume tank experiments, which could yield time series, i.e. 'snapshots' (function spaces) of evolving landscapes that could be used to learn the mapping, e.g. erosional processes and perhaps uplift histories. Similarly, advective and diffusive PDEs are widely used to generate predictions of landscape geometries (e.g. fluvial, glacial and hill slope topography) and their evolution. Function spaces (i.e. synthetic landscapes) can easily be generated from the solutions to such equations, which could be used to develop neural operators. A useful benefit of the neural operator approach is that, once the learning is done, future function spaces (maps of elevations) can be predicted very rapidly (see Section 5.5 in Li et al., 2022, for a fluid mechanics example).

Here, I focus on exploring whether such operators can be established from synthetic landscapes generated using the deterministic stream power model. This model has the form of a nonlinear advective PDE and is used to predict fluvial landscape evolution at a range of scales, from river reaches to continents. I seek to establish whether a deep learning algorithm can determine the operator required to map (convert) a stream power derived landscape from one time step to another. In turn, I want to understand if the operator can be used to reliably predict evolution of the landscape at subsequent time steps. Positive answers

to those questions would indicate that Fourier Neural Operators can be used to model landscape evolution, providing a step change in the speed at which evolution of landscapes can be computed once the operators are learned. More broadly, it would, with further work, perhaps provide new tools to generate novel insight into the processes that drive landscape evolution.

## 2   Methodology

### 2.1   Generating the training information from a LEM

The training information—a set of landscapes: $z(x,y)|_{t^*=0}, z(x,y)|_{t^*=1}, \ldots, z(x,y)|_{t^*=9}$, where elevation, $z$, is a function of spatial coordinates, $x, y$, and $t^*$ indicates time step indexing—was produced by solving the stream power model using Landlab routines (Hobley et al., 2017, see Code Availability Statement). The model solved has the form

$$\frac{\partial z}{\partial t} = -vA^m \nabla z, \tag{1}$$

where $z$ is elevation, $t$ is time and $A$ is upstream drainage area (see e.g. Lague, 2014; Hobley et al., 2017, and references therein for additional information about the stream power model and its parametrization in two dimensions). In the examples in this paper, erosional parameters $v = 0.3$ Myr$^{-1}$ and $m = 0.5$. The model domain is a $(x \times y)$ $128 \times 128$ km square with cell size, $\Delta x = \Delta y = 1$ km. The starting condition is a 1 km high block of topography across the entire domain and additional small-amplitude, uniform (white) noise, which, as is typical in such models, is included so that channel networks with realistic geometries form. All boundaries are fixed at zero elevation for the duration of the model. Figure 1 shows example output from the model for a few of the first time steps. As expected, the resultant landscape resembles four escarpments advecting headwards (upstream) from the boundaries, more or less towards the center of the square domain. The exact arrangement of the channels, including their headwaters, depends on the specific noise function (see e.g. Kwang and Parker, 2019; Morris et al., 2023). The first ten time steps ($t^* = 0, 1, \ldots, 9$; time step length $\Delta t = 1$ Ma) are used to train the Neural Operator.

### 2.1.1   A note on computational speed of existing LEMs

There are two main concerns with regards to computation time when solving the stream power model numerically, e.g. via finite-difference or finite-volume methods. First, as is typical in such numerical problems, time step length, $\Delta t$, plays an important role in determining the computational time required to generate solutions at specific (model) times, and also in their accuracy and stability. Such properties are often established by ensuring that the Courant-Fredrichs-Lewy (CFL) condition is met (see e.g. Press et al., 2007; Roberts and White, 2010). In this problem it has the form

$$\frac{|vA^m|\Delta t}{\Delta x} \leq 1. \tag{2}$$

Inserting the maximum possible value of $A$ into Equation 2 should ensure stability at all times across the entire spatial domain. As an example, if we use the maximum possible drainage area (i.e. the entire domain: 16,384 km$^2$), and use the values of the erosional parameters given above, the CFL derived $\Delta t \leq 0.026$ Myr. If we set $\Delta t = 0.02$ Myr, this forward model, using

Landlab routines, takes 24.3 s on a computer with a 2.6GHz Intel Core i7 processor to produce 50 Ma of model time, by which time the initial topographic block is almost completely eroded. In practice, fairly reliable results (i.e. demonstrable convergent landscape geometries at large scales) can be obtained (for this parametrization) even when $\Delta t = 1$ Ma if the nominally implicit Fastscape scheme is used to compute erosion, resulting in a reduced run time of 2.4 s (Braun and Willett, 2013). Nonetheless, inverse modeling of landscapes for, for instance, their uplift rate histories or erosional parameter values might require in excess of $\mathcal{O}(10^5)$ forward models runs even for a modestly sized landscape, which is a considerable computational burden (see e.g. Croissant and Braun, 2014).

Within a single time step, flow routing and calculation of upstream drainage area, $A$, from flow routing algorithms, is nearly always the slowest computation, and the second major concern. Recent advances to reduce computation time include careful parallelization; partitioning flow routing calculations to different computational nodes (Barnes, 2019). Nonetheless, it would be helpful if flow routing, and time stepping, could be avoided altogether. I now explore whether time stepping and flow routing can be avoided by making use of Neural Operators.

## 2.2 Neural Operator

A Fourier Neural Operator is used to learn the mapping between the evolving landscape at different time steps based on the approach introduced in Li et al. (2022). This deep learning approach makes use of Fourier transforms to parametrize a kernel integral operator, which is learned from the evolving landscape.

For the specific problem of interest (and often in geomorphology generally), we wish to determine the operator $G^*$ that maps (via the erosional process, say) elevations in a landscape at one time, $\mathcal{Z}_\tau$, to those at another time, $\mathcal{Z}_{\tau^*}$. Directly approximating operators, $G \approx G^*$, can be very computationally cheap and fast (Li et al., 2022). For the problem of interest, we seek to recover $G$ from synthetic landscapes at discrete time steps,

$$G : \mathcal{Z}_t \rightarrow \mathcal{Z}_{t+n}, \tag{3}$$

where $t$ and $t + n$ indicates time step indexing. In the examples examined in this paper $n = 1$, i.e. we seek to learn $G$ from landscapes at adjacent time steps. Since elevation information can usually be cast as point-wise data it is straightforward to define input-output pairs, e.g. $\mathcal{Z}_t = z(x, y)|_t$ and $\mathcal{Z}_{t+1} = z(x, y)|_{t+1}$.

The neural operator is formulated in three main steps (see Li et al., 2022, for details). First, the input data (e.g. $\mathcal{Z}_{t=0}$) is lifted to a higher dimensional representation with an encoder network (see also Kovachki et al., 2023). Secondly, four layers of integral operators and activation functions are then applied. The 'integral operators' are actually convolution operators defined in Fourier space. The scheme uses Fourier modes up to $k_{max}$, and as such acts as a low-pass filter. Finally, the output is then projected back to the target dimension by another neural network. In each iteration, the update $\mathcal{Z}_t \rightarrow Z_{t+1}$ is defined as the composition of a non-local integral operator $\mathcal{K}$ and a local, non-linear activation function, $\sigma$ (see Li et al., 2022, especially their Figure 2, for an extended explanation). A minimum working example, demonstrating how the calculations are performed is provided (see Code Availability statement for details).

An Adaptive Moment Estimation (Adam) optimizer is used to train the model, which minimizes differences between $Z_{t+1}$ and $\mathcal{Z}_{t+1}$. Thus, the operator, $G$ is defined and can now be used to generate predictions of landscape evolution at other, say, intermediate, and, perhaps more usefully, later times. Specific implementations for the examples discussed in this paper are archived (see Code Availability Statement). For the two examples shown in this paper, model A was trained for 100 *epochs*

(number of times the learning algorithm uses the entire training set), with $k_{max} = 2$. Model B was trained for 500 epochs, with $k_{max} = 4$. The initial *learning rate* (determining rate at which parameters in the model are updated) in both models was defined as $10^{-5}$, which was halved every 100 epochs. The number of training and testing sets (landscapes between 0–9 time steps; see Section 3) was held constant at 100 and 20, respectively, and model 'width', which plays a role in lifting the input to a higher dimensional representation, was fixed to be 20. All computation was performed on a single Nvidia GPU. Training

took $< 1$ hour for each model tested.

## 3   Results and Discussion

Figure 1 shows a subset of the space functions (landscapes) at time steps 0 to 9 generated by the stream power model (with $\Delta t = 1$ Ma), used to train the neural operator. The full training set is archived (please see the Code Availability Statement for details). Figure 2 shows predictions from the neural operator models A and B at time steps 10, 20, 30 and 40. For comparison,

adjacent to those predictions are the solutions to the stream model at the same times.

Topographic swaths, histograms and hypsometric curves summarizing the distribution of elevations from the three models (stream power, FNO models A and B) are shown in Figure 3. The neural operator models do a reasonably good job of capturing the large-scale structure of the evolving landscapes. Like the stream power model, they both include headward 'advection' of the four main escarpments. The FNO approach, as implemented, acts as a low-pass filter and hence fine detail, such as valley

networks, are lost. The resultant landscapes tend to be smoother when compared to predictions from the stream power model (Figure 2). They also lack the well developed channels present in landscapes predicted by the stream power model, which is unsurprising given the low pass filtering. Topographic swaths that traverse the centre of the landscapes and from corner to corner further emphasise the relative smoothness of topography predicted by the FNO models, their tendency to be similar to each other (for the model parameterisations tested), and their capture of the overall lowering of topography (Figure 3). Given

the filtering, it is not surprising that the FNO models predict greater or less local (in space and time) erosion than the SPM model. Nonetheless, as the histograms and hypsometric curves in Figure 3 indicate, changes (here only reductions) in elevation across the domain are faithfully reproduced. Increasing the number of epochs increases the presence of short wavelength structure in landscapes predicted by the neural operators. However, doing so can lead to development of local patches of noisy and negative topography (see black pixels in Figure 2i and 2l).

Unsurprisingly, the rougher parts of the FNO landscapes (towards the centre of the domain) tend to have more sinks (local depressions). Consequently, drainage in the FNO landscapes tends not to be connected and through-going in upper reaches when calculated using the widely used D8 flow routing algorithm (using the Flow Accumulator Landlab component; Tarboton, 1997; Hobley et al., 2017). These results are analogous to what happens to a SPM when noise is inserted—a generally accepted

step that enables realistic planforms to emerge once sink-filling has been performed (see e.g. Barnes et al., 2021, for a useful summary of sink-filling approaches). I note that sink-filled versions of the two FNO landscapes examined have connected drainage with planforms that are broadly consistent with those predicted atop equivalent SPM landscapes. It is an interesting and probably quite fundamental question whether we should regard the presence of sinks as being important or not when we no longer need a flow routing algorithm to evolve a landscape? A key question is what do we want these models to predict? For instance, we currently sacrifice realism in both the SPM and FNO approaches at specific scales to obtain solutions of interest. Perhaps, it is encouraging that the FNO models can produce reasonable predictions of landscape geometries at large scales even when channel elevations are not monotonic. It may be interesting to explore the use of FNOs to learn landscape evolution from landscape evolution models that more explicitly incorporate physics and generate realistic geometries across the scales of interest, perhaps one way to do so is to make use of landscapes predicted by stochastic theory that naturally includes/can cope with the presence of sinks (see e.g. Roberts and Wani, 2024, and references therein).

These results suggest that erosion, at least at large scales, can be learned from evolving landscapes even when the erosional laws are quite complex, depending on, for instance, advective velocities that depend non-linearly on upstream drainage area. In turn, the learned operators can be used to predict landscape evolution at relatively large temporal scales. These results suggest that use of such an approach in the development of inverse methodologies that seek to calculate uplift or denudation histories from observed fluvial landscapes could be fruitful. For instance, they might find use in developing understanding of landforms generated in response to tectonic or sub-plate processes. Such techniques might find use in examining domal topographic swells and continental escarpments, for example, where the specific details of geomorphic geometries (e.g. historic channel locations) are perhaps less crucial (or knowable) than the larger scale changes in landform geometries (see e.g. Roberts and White, 2010; O'Malley et al., 2021). This approach could be particularly useful when the objective functions used to minimize misfit between observed and theoretical landscape are designed to 'see through' the impact of local noise; when specific positions of calculated channels and interfluves are largely unimportant (see e.g. the Wasserstein based approach introduced in Morris et al., 2023).

In may be fruitful to explore the use of alternative training information to develop useful operators, for instance, it would be straightforward to develop training data (input-output pairs) using maps of upstream drainage areas or sedimentary flux predicted by SPMs. Or perhaps useful operators could be generated from solutions to different PDEs or stochastic theory (e.g. Equation 4 in Bonetti et al., 2020; Roberts and Wani, 2024). An obvious concern is that the functions used to generate the operators are sensitive to the things we want to know about. For instance, using elevation appeals to me because of my interests in solving for uplift rate histories. In contrast upstream drainage areas, at least at large scales, appear to be quite insensitive to some uplift rate histories, e.g. the extreme examples of uplift only varying as a function of time or very smoothly as a function of space (see e.g. Roberts and White, 2010; O'Malley et al., 2021). The general theme of structure, interconnectedness, and accuracy of FNO derived geometries are probably worth examining further in future work.

Clearly, establishing whether FNOs (or other deep learning approaches) developed for one environment, or set of model parametrizations, can be ported to predict landscape evolution in other settings (e.g. driven by different uplift histories or erosional forcings) is likely to be important future work. Evidence from other domains is promising (see e.g. the results in

Li et al., 2022). More work is also required to establish optimal model parametrizations, e.g. numbers of testing and training sets, epochs, learning rates. I note that Li et al. (2022) and Kovachki et al. (2023) discuss how, despite truncation of higher frequency modes in the Fourier layer, the operators they produced, as a whole, approximated the functions they examined (e.g. solutions to the Navier-Stokes equation) to frequencies considerably higher than $k_{max}$ with low error. Li et al. (2022), in their response to reviewers' comments, attribute those results to the lifting of input functions to higher dimensional representations. It will be interesting to establish whether increasing the dimensionality of input landscapes beyond what is explored in this paper ('width' = 20; see Code Availability statement) improves predictions of fine structure (e.g. valleys and interfluves), or whether landscapes are special in some way (e.g. perhaps because of flow routing). Preliminary results indicate that changing 'widths' between 1 to 32 (whilst holding all other parameters constant) has little impact on predicted landscapes.

Despite the work to be done, it seems clear at this early stage that there can be benefits to using FNOs, including the fact that once an operator has been generated, predicting landscapes (the 'forward model') is much more efficient than solving the partial differential equations numerically. Such an approach facilitates efficient parameter sweeping and 'filling-in' gaps between time steps, for instance. More broadly, it seems likely that development of operators from 'analogue' landscapes generated in flume tanks or perhaps from repeat topographic surveying could provide means to develop new understanding of the processes at play in evolving landscapes.

## 4 Conclusions

This paper introduces the use of Fourier Neural Operators (FNOs) for predicting evolution of landscapes. This deep learning methodology was trained using a simple synthetic landscape that was evolved forward in time using the well known stream power erosional model. This deterministic kinematic model advects slopes headwards with velocities that depend on upstream drainage area and defined values of erosional parameters. Time steps 0 to 9 were used to generate 'learning maps' between the function spaces (landscapes). The learned operators were then applied to predict landscape geometries at time steps 10 to 40. The resultant landscapes were compared to solutions from the stream power model. Two different FNO parametrizations were tested with different Fourier mode filters and learning epochs. Both reproduce solutions from the stream power model at large scales. These results indicate that developing FNOs for landscapes might be a fruitful way to increase the speed with which landscape evolution can be modeled and generate new understanding of erosional processes. An important piece of work to be done is developing understanding of whether operators developed using observations or model output from one setting can be ported to understand landscape evolution in other contexts.

*Code availability.* Code, parametrization files and example output used to generate the training information and digital elevation models for validation, code used to generate the Fourier Neural Operator (FNO) and ancillary implementation and plotting scripts, which contain information about how to run the code on a GPU system and how to manage the resultant `.mat` files are archived with doi: 10.5281/zenodo.14616760. Note that the material used to develop and run FNOs is based on work from Li et al. (2022) and Kovachki et al. (2023).

*Author contributions.* GGR: conceptualisation, coding, analysis, paper writing.

*Competing interests.* None.

*Acknowledgements.* I thank Francois van Schalkwyk and Matthew Morris for their help. I also thank Simon Mudd, Christoph Glotzbach and an anonymous referee for their suggested improvements and broader commentary.

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

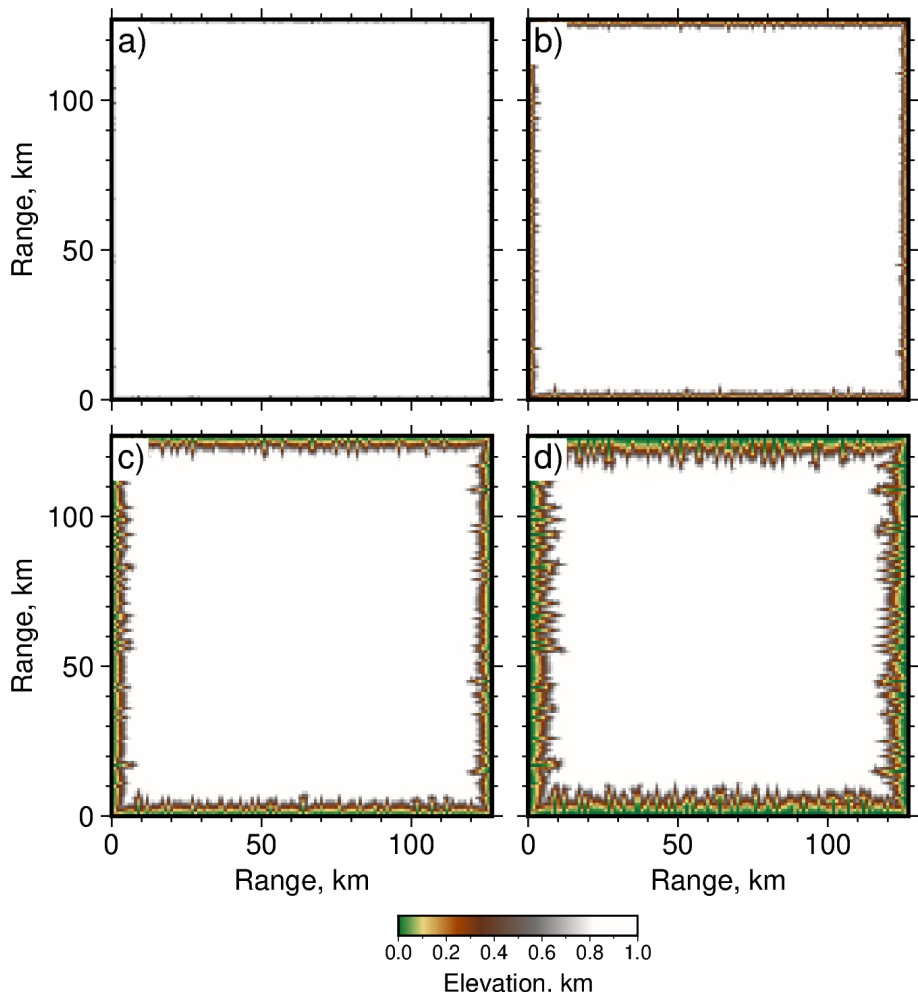

**Figure 1. Evolution of the synthetic landscape used to train the Neural Operator.** The entire training set incorporates digital elevation models at ten time steps ($t^* = 0, 1, \ldots, 9$, $\Delta T = 1$ Myr), generated by solving Equation 1. Examples of the training 'function spaces' (i.e. digital elevation models) at time steps (a) 0, (b) 3, (c) 6 and (d) 9 are shown. The domain is $128 \times 128$ with grid resolution = 1 km (16,384 cells in total).

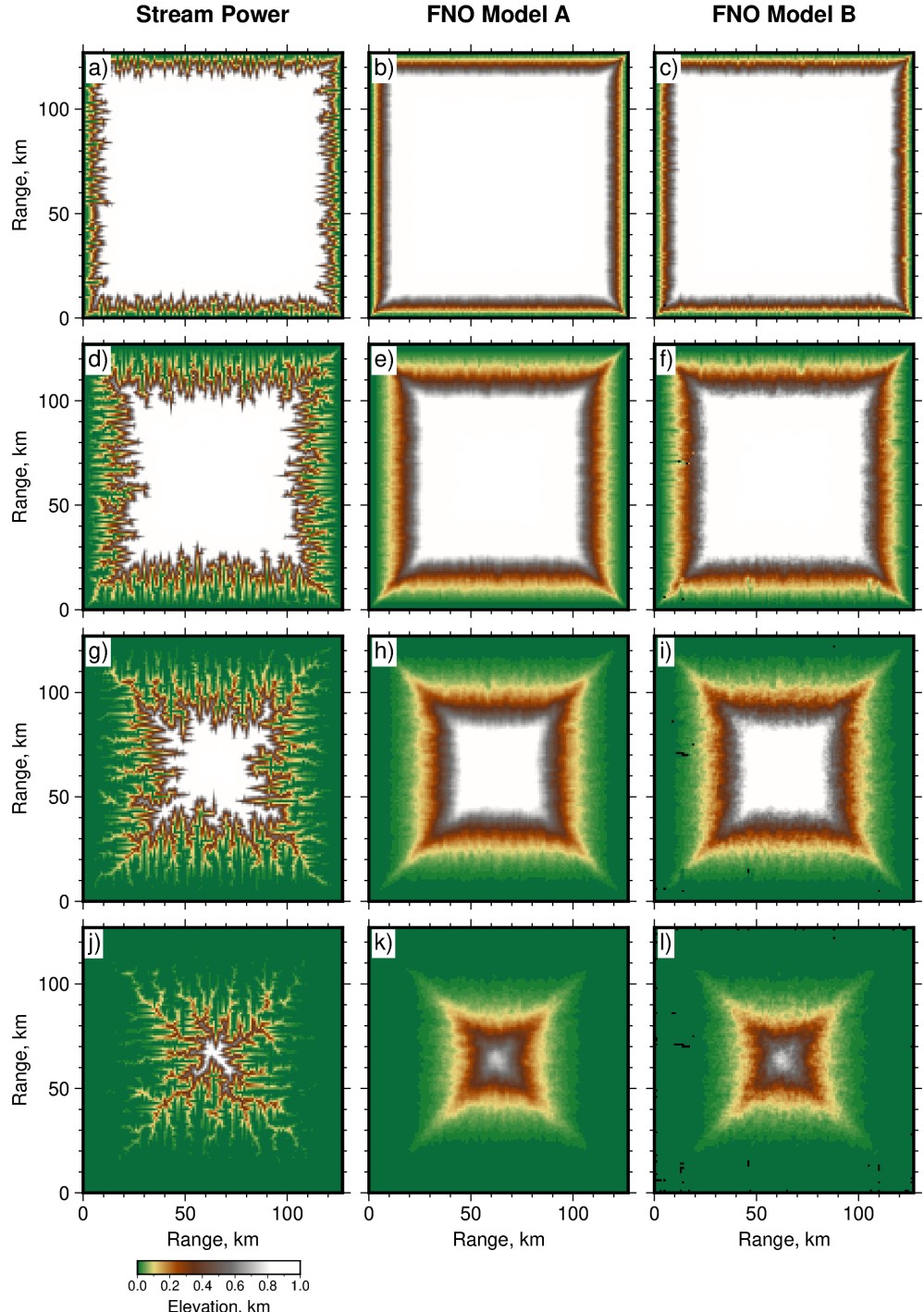

**Figure 2. Predicted landscape evolution from stream power and Fourier Neural Operator (FNO) models.** (a-c) Predicted landscapes at time step 10 from (a) the stream power model, (b) FNO model A, (c) FNO model B. (d–f), (g–i), (j–l) Predicted landscapes at time steps 20, 30 and 40, respectively. See body text for FNO model parametrizations.

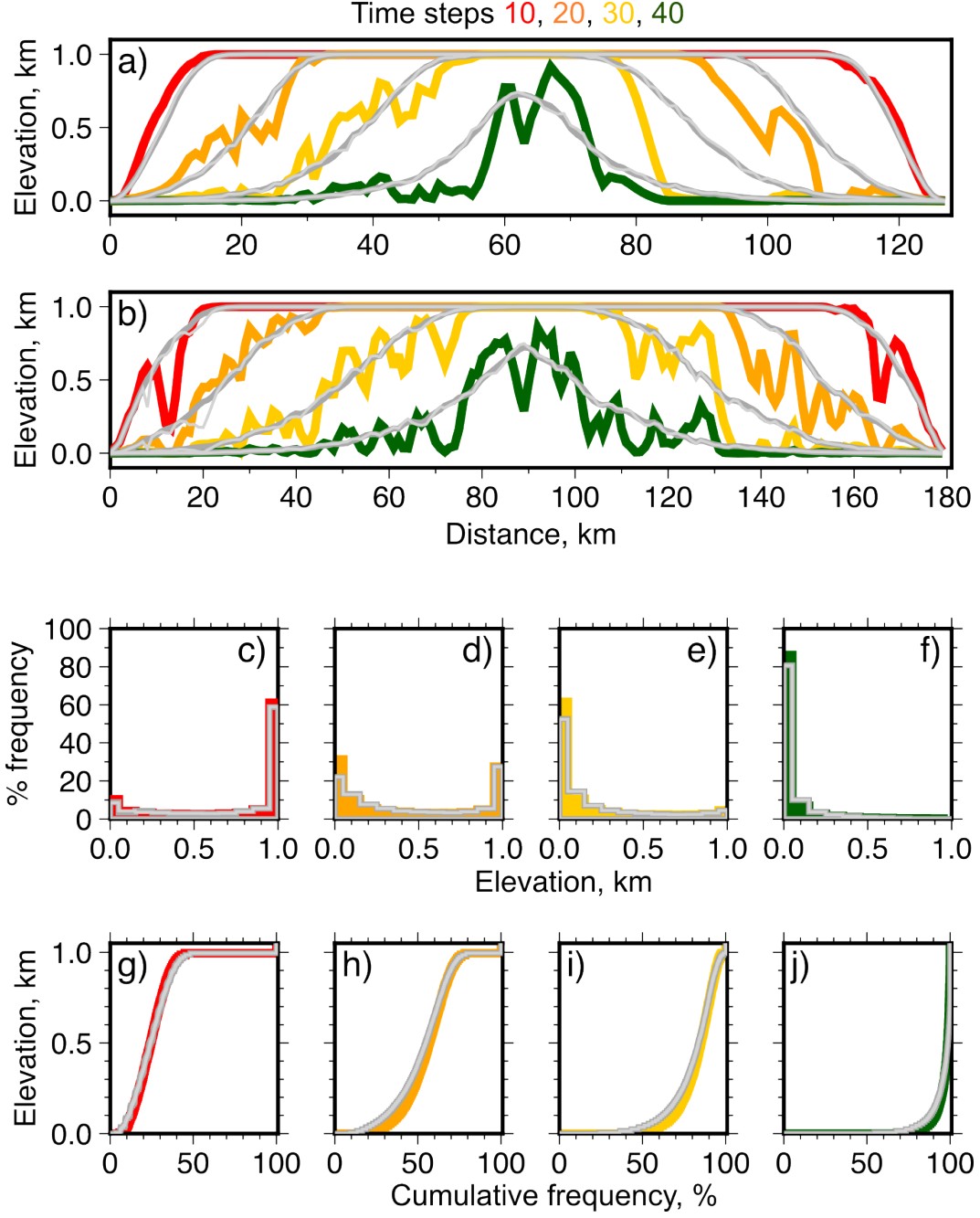

**Figure 3. Comparison of topographic transects and histograms from evolving landscapes generated with stream power (SPM) and Fourier Neural Operator (FNO) models.** (a) Topographic transects across the landscapes shown in Figure 2 from coordinates (0,64) to (128,64), i.e. 'west' to 'east'. Thick coloured lines = transects through SPM model at annotated time steps; dark and light grey lines = transects though FNO models A and B at those time steps, respectively. (b) Transects from bottom left to top right corners—(0,0) to (128,128)—across the landscapes shown in Figure 2. (c–f) Percentage frequency histograms of elevations shown in Figure 2, and (g-j) associated hypsometric curves; line stylings as for panel (a).