# Peer review of "Short communication: Learning How Landscapes Evolve with Neural Operators"

_EGUsphere, 2025_

## Author Comment (AC1)

**Response to comments on manuscript entitled "Short communication: Learning How Landscapes Evolve with Neural Operators"**

**Response to comments from RC1**

**The manuscript applies a deep learning method called Neural Operators or more specifically, Fourier Neural Operators to investigate if they can accurately predict landscape evolution in models produced with the stream power model. Results indicate that the first-order landscape form (e.g. hypsometry) can be reliable and rapid predicted, while fine-scale features (e.g. river valleys) are not predicted. Although the results are promising, more work is required to evaluate the applicability of the approach, and I would like the author to state the limitations and need for further detailed analyses in the abstract and conclusion.**

I thank the referee for their assessment of this paper and suggested improvements. In short, I agree with them that more work is required to evaluate the general applicability of FNOs to landscape evolution problems. I have emphasised the current limitations and need for further analyses in the revised abstract and conclusions. I respond to their detailed comments as follows.

**Response to "Detailed comments"**

**Line 1-8: You may want to add the limitations at the end of the abstract, e.g. that it is not yet explored if learned operators can be applied to other parameterisations.**

I have now inserted '*interesting future work could involve assessment of whether learned operators can be applied to other settings or model parametrisations*' into the abstract.

**Line 74: Please clarify how a time step of even 1 Myr is possible, although the condition in Eq. 2 is not met.**

The landscape evolution model will run when the time step is 1 Myr (or even larger), but of course stability and accuracy of solutions (in general) becomes a concern when the Courant condition (Equation 2) is not being met. The main reason why I think we can 'get away with' using a timestep of 1 Myr here is because a nominally implicit scheme is used to compute the erosion (Fastscape; Braun & Willet, 2013).

**Line 117: I guess with epoch you mean time steps. Might be good to use through the paper either time step or epoch.**

I have clarified in the revised manuscript '…*epochs* (number of times the learning algorithm uses the entire training set)…'.

**Line 118: Please define 'learning rate' in the previous paragraph.**

Done as suggested. I now include `…*learning rate* (determining rate at which parameters in the model are updated) …'.

**Line 119: Can you explain in more detail what you mean with training and testing sets.**

Done. I now include '…training and testing sets (landscapes between 0—9 time steps; see Section 3)…'.

**Line 144: Since one of the starting objectives is to find a faster alternative to conventional LEM, it is quite important to state if the trained operators can be used to run other scenarios, e.g. with different boundary conditions and parameterisation. Not much is gained if training needs to be repeated for each individual scenario solved, for instance in an inverse approach. Please state your experiences**

I agree that examining whether FNOs (or other *deep learning* approaches) developed for one environment (or set of model parametrizations) can be ported to predict landscape evolution in other settings (e.g. driven by different uplift histories or erosional forcings) is an important question. The evidence from other domains (see e.g. the results in Li et al. 2022, discussed in the main manuscript) is promising. However, I'm afraid that I don't (yet) have much domain-specific experience to report.

I hope that this paper helps to spur work in that direction. Nonetheless, even if porting is not possible, I think there may be utility in the approach for other problems. For instance, it might prove useful for developing understanding of rules governing landscape evolution in analogue (e.g. flume tank experiments) or for 'filling in gaps' between time steps in numerical experiments.

**Line 148-150: This might be a good application and would save not computational time but time in the lab. Might be worth mentioning in the conclusion/abstract.**
Thank you for the suggestion. I have added a statement to the abstract "…*neural operators could be used to learn the processes that evolve actual and analogue landscapes*", which I hope is satisfactory.

**Line 159: Please mention that the computational time can only be reduced in case the produced operators are applicable/scalable to other model parameterisations.**
Thank you. To address this comment I have revised the conclusions to include "*these results indicate that developing FNOs for landscapes might be a fruitful way to increase the speed with which landscape evolution can be modeled and generate new understanding of erosional processes. An important piece of work to be done is developing understanding of whether operators developed using observations or model output from one setting can be ported to understand landscape evolution in other contexts*".

**Equations:**
**Eq. 3: You may want to add spatial coordinates *x,y* to *G, Z*.**
I would prefer not to add *x* and *y* notation to *G* and *Z* in Equation 3 because I feel they would complicate the main message here, which is that we are attempting to produce an operator from landscapes at different times. The spatial coordinates used to define *Z* in Equation 3 are described in the second sentence that follows the equation, which I hope is satisfactory.

**Figures:**
**Fig. 1 Caption: Change to '…used to train the…'**
Thank you! Done as suggested.

**Fig. 1 Caption: Add the used time step, I guess 1 Ma.**
Done as suggested.

**Response to comments from RC2**

**This manuscript briefly illustrates how a method from the deep learning community might be used as a computationally efficient emulator for numerical solutions to fluvial landscape evolution equations. Having such an emulator, especially one that is independent of grid resolution or particular parameter values, would be a big advantage in applications where a great many model solutions are needed (for example, in studies that use optimization to infer parameter values or boundary conditions from digital elevation data). A strength of the manuscript is that it introduces readers to a promising technique from a different field, and demonstrates that it has some potential for geomorphology. Another strength is that the author's code and related files are available online for anyone to try out.**
Thank you for the encouragement.

**One limitation of the manuscript is that it does not provide details of the methodology, referring readers instead to a conference paper by Li et al. (2022). That paper introduces a deep learning approach for approximating the solution of PDEs, but it is geared toward an audience versed in the relevant applied mathematics; to understand it fully would probably require a fairly major effort for most geoscientists. Given that the author has put in the work**

**to understand and apply the method of Li et al. (2022), it would add value to the manuscript if it were to provide a more extensive and practical translation for eSurf readers: a geoscience-friendly explanation of this novel method from the deep learning world.**

Thank you for the suggestion. I have revised (expanded) the description of the methodology, which hopefully now provides a clearer, more practical, introduce to the approach. Please see the Tracked Changes document for specific changes.

**The manuscript demonstrates nicely that the FNO can capture the broad-wavelength pattern of terrain evolution. One shortcoming, which the manuscript acknowledges, is that the FNO loses the details of the valley networks. It would be interesting to see an interpretation of why this is the case. The manuscript notes that the Fourier operator acts as a low-pass filter. Could this be why the FNO models do not capture valley network features? It would be interesting also to know whether the FNO models can produce topography that drains (i.e., does not contain spurious internally drained basins).**

I think the referee's view is correct: the FNO approach, as implemented, acts as a low-pass filter and hence fine detail, such as valley networks, are (purposefully) lost. I have revised the body text to address this comment, it now includes "*they also lack the well developed channels present in landscapes predicted by the stream power model, which is unsurprising given the low pass filtering*". With regards to the referee's second point, please see the figures in this document. Figure 1 shows the result of applying a D8 flow routing algorithm without filling of sinks to the FNO and stream power model (SPM) derived landscapes at time step 40 (shown in Figure 2m-o of the main manuscript). Figure 2 shows results when sinks are filled. Both were produced using the *FlowAccumulator* Landlab component. For clarity, only the main channels of the 100 largest 'rivers' (by area) are shown. Unsurprisingly, the rougher parts of the FNO landscapes (towards the centre of the domain) tend to have more sinks ('*Flows to Self*' nodes in Landlab parlance), resulting in less connected, throughgoing, 'drainage' in upper reaches. I think it is an interesting and quite fundamental question whether we should regard the sinks (and resultant disconnected drainage patterns in headwaters) as being spurious or not. Sink filling is of course well established in stream power modelling. However, it is also generally accepted that noise should be inserted into the models to enable realistic planforms to emerge. The consequence of inserting noise into a SPM is disconnected drainage (unless sink filling is performed). In the end, I think the key question is what do we want these models to do/produce? We currently sacrifice realism (in both the SPM and FNO apporaches) at specific scales to obtain solutions of interest/use. Perhaps, it is encouraging that the FNO models can produce reasonable predictions of landscape geometries at large scales even when channel elevations are not monotonic. The sink-filled FNO landscapes have connected drainage (as expected; Figure 2 of this document). Note that I have not included these results in the main manuscript because I think they are somewhat tangential to the main goals here. Nonetheless, I think the general theme of structure, interconnectedness, and accuracy of FNO derived geometries are worth examining further in future work.

[Figure]

Figure 1. Same annotation as Figure 2m-o in the main manuscript with addition of black curves showing the results from flow accumulation without filling of sinks. 100 largest channels (based on drainage area at the perimeter) are shown.

[Figure]

Figure 2. Same annotation as Figure 1 in this document but with flow accumulation performed with sink filling.

**One question I had in reading the manuscript was whether the FNO might have better success if it were trained not just on topography but also on contributing drainage area. The text after line 90 refers to the calculation of A, which is presumably done by a routing algorithm rather than by a traditional solution to a PDE. Bonetti et al. (2020) pointed out that the calculation of A (in the form of specific contributing area, a) can be cast the solution to the PDE:**
**-div (a grad(z) / |grad(z)|) = 1**
I think that this idea is very interesting and well worth further investigation. As you say, upstream area (in the SPM model) was calculated using a flow routing algorithm (D8). So long as we can define input-output pairs, i.e. $f_1(x,y)$ and $f_2(x,y)$, which is straightforward for A, and I think possible for the PDE you describe, it would be quite straightforward to insert them into the existing code to assess whether the resultant operators provide useful predictions. An obvious concern is that the functions used to generate the operators are sensitive to the things we want to know about. For instance, using elevation appeals to me because of my interests in solving for uplift rate histories. Upstream area can be quite insensitive to some uplift rate histories, e.g. the extreme examples of uplift only varying as a function of time or very smoothly as a function of space. Nonetheless, I think the idea of examining the use of other landscape-derived functions is a great one.

**Viewed from this perspective, the numerical model in Figure 1 can be thought of as solving TWO coupled PDEs: the erosion law of equation (1), and the equation that governs A (or a). These two PDEs in combination produce the time-evolving fields of z and A. Perhaps the loss of valley features reflects training the FNO models only on z and not on a? Admittedly, testing this would probably involve considerably more work and additional material in the manuscript, which might be beyond the scope. But it does seem worth considering, either for this piece or for a possible follow-up one.**
I agree! If we can define function spaces that incorporate elevations and areas explicitly then I think it would be very straightforward to experiment with them using the existing code. However, apart from that very broad statement, I can't provide any more insight at this stage. I agree that it would be interesting to follow up on it.

**The manuscript notes that 'A useful benefit of the neural operator approach is that, once the learning is done, future function spaces can be predicted very rapidly'. One minor suggestion is to explain what 'function spaces' means (though it's easy enough to guess that it refers to future values of the dependent variable z). More broadly, the Li paper argues that an**

**advantage of the FNO approach is that it is not restricted to a particular numerical discretization or a particular set of parameter values. In other words, the FNO 'learns' the PDE itself somehow. If this is right, then one suggestion for the manuscript is to demonstrate this advantage by trying out models that use a different discretization or a different value of v from the training data.**

Thank you. With regard to the referee's first point, I have modified this part of the manuscript to "…*once the learning is done, future function spaces (maps of elevations)*…". I am in complete agreement on their second point, but think that it is work for a future paper.

---

## Author Response (AR2)

**Responses to comments from Associate Editor on manuscript entitled 'Short communication: Learning How Landscapes Evolve with Neural Operators'**

**I've now looked through the reviewer comments, responses, and revised manuscript. Both reviewers found the approach quite interesting, were generally positive about the paper, and asked for some clarifications and more discussion. The revision contains some of this requested material, but in some places the review and response is much more enlightening than the final manuscript. I suggest bringing more of that discussion into the paper.**

Thank you for the guidance. I have moved much of the discussion that featured in the review and response document into the main manuscript, with some modification to aid flow. Details on what was incorporated, and in some places expanded upon, follows. The main changes are to the discussion of the methodology and the results and discussion section.

**There were requests to clarify what the technique does, and I think a few extra statements do a good job of making the approach easier to understand. However, there is a section where the mechanics of the technique are explained, this is reduced to five or six sentences, and an encouragement to read the Li paper. I think this section could add a few more sentences to help readers (i.e. a bit more on the steps of increasing then decreasing the dimensionality with the neural network). I leave it up to the author if they think clarifying details can be added succinctly.**

I have added more detail on the Fourier Neural Operator approach, including its parameterisation in Section 2.2. This new material gives (a) additional information about how the input data is raised to a higher dimensional representation, following the procedures described in Li et al., 2022, (b) parameterizations used to do so in this study, i.e., the values of model 'width', (c) reference to Kovachki et al. (2023), where the apparent benefits of lifting the data to higher dimensions is discussed, and (d) a sentence directing the reader to the code provided, which includes the lift and later reduction in dimensions. In the expanded results and discussion section I return to the topic of raising to a higher dimension. In there I briefly summarise the responses that Li et al. provided to reviews and those that Kovachki et al. discuss in their paper regarding their results. I summarise their demonstrations of spectra at frequencies higher than $k_{max}$ being reliably predicted despite higher Fourier modes being removed when generating the operators. They attribute these results to the raising of input data to higher dimensions. I give a (brief and tentative) view on why flow routing and the importance of noise in landscape evolution models might make achieving similar success, e.g. accurate prediction of specific valley geometries, challenging for landscapes. This discussion includes a summary of preliminary tests I have run, which demonstrate that changing model 'widths' from 1—32 appears to have little impact on the geometries of predicted landscapes. I note here, and in the revised manuscript, that these results are very much preliminary, and that further testing, e.g. lifting to much higher dimensions, might be fruitful.

**A bigger issue for me is the reviewer comment about flow routing, valley structure, and connectivity. There is an interesting discussion in the response that does not find its way into the paper. The predicted surfaces do not have the same obvious features as the stream power model results. They lack the ridge and valley structure and degree of dissection in the stream power model**

**outputs. This is attributed to the smoothing effect of the fourier convolution. In addition the networks are not connected. The stream power model forces pit filling and drainage connectivity and the FNO appears unable to capture this behaviour. The reviewer suggests a workaround, which is then addressed in the response letter. I strongly urge the author to add more discussion of this in the manuscript.**

I have expanded the discussion of sinks and flow routing in the results and discussion section to now include the material on lines 145 to 159, which I hope is satisfactory.

**In addition, we are shown a comparison of the stream power-derived surface and FNO surfaces using a very basic elevation binning that only includes 10 bins. It is a rather crude approximation. Can I suggest adding in the hypsometric curve (which will give better granularity on where the models differ) and I think it would be quite interesting to have some swath profiles coming in from the boundary of the two sets of surfaces. The upstream tip of the erosion wave appears to be well reproduced, but how well does it do with the average mass loss behind that tip? I think a swath would show that quite clearly.**

Thank you for the guidance, which I have followed to produce the new Figure 3. It shows transects through the SPM and FNO models, the histograms that were in figure 2 and the new hypsometric curves. I see little evidence in these panels, nor in the revised Figure 2 of a systematic bias in the predictions from the FNO models, e.g. erosion of the 'escarpments' being too quick or slow. Instead, I think the FNO models are providing 'smooth' predictions of landscape evolution, looking like a low pass filter of the predictions from the SPM model.

**Overall I think this manuscript does an excellent job of introducing a new method to the landscape evolution community, it shows that the method is ripe for more exploration, and I will be happy to see this in ESURF with a few cosmetic changes.**

Thank you. I hope that the new changes are satisfactory.